# The IgSF Cell Adhesion Protein CLMP and Congenital Short Bowel Syndrome (CSBS)

**DOI:** 10.3390/ijms24065719

**Published:** 2023-03-16

**Authors:** Fritz G. Rathjen, René Jüttner

**Affiliations:** Max-Delbrück-Center for Molecular Medicine, Robert-Rössle-Str. 10, 13092 Berlin, Germany; r.juettner@mdc-berlin.de

**Keywords:** CSBS, CLMP, intestine, malrotation, smooth muscle cells, cell adhesion, filamin A

## Abstract

The immunoglobulin-like cell adhesion molecule CLMP is a member of the CAR family of cell adhesion proteins and is implicated in human congenital short-bowel syndrome (CSBS). CSBS is a rare but very severe disease for which no cure is currently available. In this review, we compare data from human CSBS patients and a mouse knockout model. These data indicate that CSBS is characterized by a defect in intestinal elongation during embryonic development and impaired peristalsis. The latter is driven by uncoordinated calcium signaling via gap junctions, which is linked to a reduction in connexin43 and 45 levels in the circumferential smooth muscle layer of the intestine. Furthermore, we discuss how mutations in the *CLMP* gene affect other organs and tissues, including the ureter. Here, the absence of CLMP produces a severe bilateral hydronephrosis—also caused by a reduced level of connexin43 and associated uncoordinated calcium signaling via gap junctions.

## 1. Introduction: Characteristics of CSBS

Congenital short-bowel syndrome (CSBS) is a rare and severe human gastrointestinal disorder that presents in newborns and is characterized by several functional alterations (OMIM 615237). The incidence of CSBS is thought to be less than one in one million births (orpha.net). No pharmacological treatments are currently available. It was initially described by Hamilton et al. [1], and its clinical appearance was recently reviewed by Van der Werf et al. [2] and Negri et al. [3]. In this review article, we concentrate on the genetic basis for CSBS and discuss a role for the cell adhesion protein CLMP (CAR-like membrane protein) in the disease. Although the function of CLMP is not fully understood, studies using a *Clmp*-null murine model have shed some light on the molecular changes that occur in the intestine in the absence of the protein. The most prominent of these are a reduction in the levels of connexin43 and 45 and uncoordinated calcium signaling via gap junctions. In addition, we discuss a putative link between CLMP and the actin-binding protein filamin A (FLNA), as mutations in this gene are also known to impair the function of the intestine.

Most CSBS patients are identified following hospitalization with bilious vomiting, diarrhea, and poor weight gain, and are diagnosed by radiological examination and exploratory laparotomy. They have a dramatically shortened small intestine with a length of approximately 50–70 cm at birth compared to 190–280 cm in healthy babies. The disease is further clinically characterized by severe malnutrition due to a decreased absorptive capacity, severe delay to thrive, poor weight gain, dilatation of the intestine, diarrhea, and bilious vomiting. Malrotation of the intestine—a broad term that encompasses several rotational and fixation abnormalities of the intestines, including mis-positionings—is another hallmark in most case reports of CSBS [2,3]. Some CSBS patients also show reduced intestinal motility [4,5] and intestinal atresia [6]. CSBS patients typically have a very poor prognosis, and despite longer parental nutrition, most die of starvation or sepsis [7]. However, some long-term survivors of CSBS have been reported [8,9], and one patient was evaluated with a good clinical prognosis [10]. In cases where intestinal biopsies were taken, standard histological staining methods revealed no obvious changes in the intestinal wall.

It has taken more than 40 years since the disease was first described, but recent studies on a small cohort of five patients from apparently unrelated families have identified a putative genetic underpinning for this severe disease. In 2012, Van der Werf et al., were the first to describe homozygous and compound heterozygous mutations in the human gene *CLMP*, which correlated with the disease in an autosomal recessive manner [11]. As has been the case with many other diseases, these observations will likely prove to be a game changer for our understanding of CSBS and the future development of pharmacological treatments.

Following the work of Van der Werf et al., there have been further case reports of CSBS patients with mutations in the human *CLMP* gene (nonsense and missense mutations, splicing errors and deletions of larger parts of the gene) [4,5,6,10,12,13,14]. The positions of these CSBS-associated mutations in relation to the structural domains of *CLMP* are shown in Figure 1. Mutations affect both Ig-like domains. Furthermore, a *Clmp* knockout mouse model and morpholino-induced knockdowns of *clmp* in zebrafish exhibit phenotypic similarities to the human disease [11,15]. Although a relatively small number of CSBS patients have been reported with mutations in the *CLMP* gene, these reports strongly implicate CLMP in this disease.

## 2. Characteristics of the Homophilic Cell Adhesion Protein CLMP—A Member of the Immunoglobulin Superfamily (IgSF)

The cell adhesion protein CLMP was discovered in 2004 in the laboratory of Jonas Fuxe by bioinformatic screens of expressed sequence tags (EST) [16]. As indicated by its name, it is highly related to the immunoglobulin cell adhesion molecule (IgCAM) CAR (coxsackie- and adenovirus receptor), with 49% similarity and 31% identity at the amino acid level. CLMP was also independently identified in a screen for genes upregulated in visceral adipose tissue in a rat model of type 2 diabetes and termed adipocyte cell adhesion molecule (ACAM) [17]. CLMP is a type I transmembrane protein composed of a signal peptide (17 amino acid residues long in mice, 18 residues in humans), a V-type (variable) and C2-type (constant) Ig-like domain, a single transmembrane segment (amino acid residues 228–250 in mice, 229–251 in humans), and a cytoplasmic tail (118 amino acid residues) with an unusual rich stretch of serine residues. A specific characteristic of the C2 domain is the presence of four cysteine residues, which form two intradomain disulfide bridges. CLMP is 373 amino acid residues long (in humans and mice) with a predicted molecular mass of 41 kD. The two Ig-like domains of CLMP contain two predicted N-glycosylation sites and CLMP runs on SDS-PAGE as a doublet with molecular masses of 45 and 48 kDa [15,16,17]. The two bands might represent different glycosylation forms. The *CLMP* gene is localized to chromosome 11q24.1 (123,069,872–123,195,248) in humans and 9 A5.1 (40,596,975–40,696,849) in mice. The genomic organization, consisting of 7 exons, is conserved between mice and humans. Exon 7 encodes the cytoplasmic segment and contains 3′-untranslated regions. So far, no other splice variants than the canonical transcript have been described.

## 3. A Survey of Mutations in the Human *CLMP* Gene Linked to CSBS and Their Putative Functional Consequences

Due to the small number of CSBS patients and a scarcity of clinical information for most individuals, a detailed phenotype-to-genotype comparison is lacking. However, missense, nonsense, splicing mutations and larger deletions in the region of *CLMP* encoding the extracellular domain have been found in association with CSBS (Figure 1). The majority of these are nonsense mutations and splicing errors of the premature mRNA, which result in a premature stop of the CLMP polypeptide and eliminate its cell surface anchorage [4,5,6,10,11,12]. Whether these truncated forms of CLMP are secreted, intracellularly degraded or whether the process of nonsense-mediated mRNA decay eliminates these transcripts with a premature stop is not yet known [18,19]. To date, no CSBS-linked mutations have been found in the cytoplasmic segment. The larger deletions in the *CLMP* gene—deletions in exons 3 to 5 with sizes of 1629, 4227, or 12,483 bp [11,12,13,14] and a deletion in intron 1 and exon 2 [11]—are unlikely to produce protein fragments. In addition, mutations that affect the splicing of the pre-mRNA are found in several introns [4,5,10,11,13]. Splicing mutations involve intronic sequences, and their precise effect on protein structure might be variable and needs to be tested experimentally.

One might expect that the missense mutations in the *CLMP* gene would exert a subtle effect on the overall protein structure, but those that have been identified prove disastrous for patients [5,11,13]. Some of these mutant CLMP proteins do not reach the cell surface and remain stuck intracellularly, as observed when the V124D or C137Y mutants are expressed in heterologous cells [5,11,20]. Others might interfere with CLMP’s capacity for homophilic binding. The impairment of proper protein folding might result in early degradation of the emerging polypeptide. This may be addressed in future studies with the generation of mouse models that carry the mutations found in human patients.

To better understand how the human missense mutations are linked to the disease phenotype, it will be necessary to assess their influence on the tertiary structure of CLMP’s extracellular region. Unfortunately, this structure has not yet been determined. However, homology modeling can be performed using the highly-related CAR protein, whose structure was solved by X-ray crystallography at a 2.19 Å resolution [21,22]. Calculations of the CLMP tertiary structure based on the murine CAR template indicate that the two proteins exhibit similar arrangements of β-sheets in their Ig domains, with a similar mode of binding between the two V-type domains (Figure 2). The two Ig domains of a CLMP monomer associate in a head-to-tail manner forming an elongated, rod-like structure. Similar to CAR, the ectodomain of CLMP may have a dumbbell-shaped surface with protrusions formed by the globular Ig domains, themselves connected by a short junction (amino acid residues VLV). Two CLMP molecules might—as described for CAR—form a U-shaped homodimer through the binding of their N-terminally located Ig domains. In murine CAR, the dimer interface for homophilic interaction has a size of 684 Å^2^ per monomer and is located at a distal part of the V domain, opposite the V-C2 junction. The two missense mutations (R69A and V124D) thus far detected in the N-terminal Ig-like domain of CLMP are located directly at the interface required for homophilic binding. Therefore, both mutations could interfere with CLMP’s ability to mediate homophilic binding between opposing cells. However, this must be tested experimentally.

The distal C2-type domain of CLMP might be arranged as a β-sandwich, with its two β-sheets formed by only three antiparallel β-strands. The overall fold might be similar to that of C-type Ig domains [23]. Two disulfide bonds link the sheets, connecting β-strand A to G and β-strand B to F. These two disulfide bonds would create a rigid structure that might also stabilize the N-terminal V domain for homophilic binding. The two missense mutations found in the C2-type domain (C137Y and C219G) eliminate both of these intrachain disulfide bonds [5,13]. This could reduce stiffness in this domain, which in turn might also interfere with the stability of the N-terminal Ig domain and affect CLMP homophilic binding.

**Figure 2 ijms-24-05719-f002:**
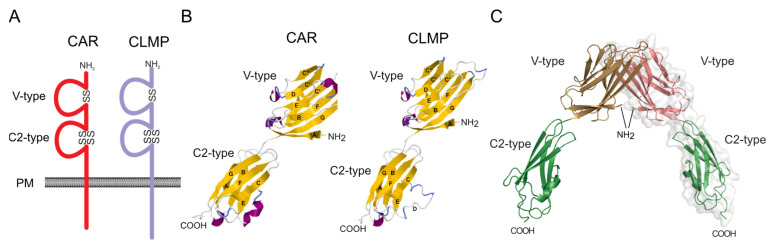
(**A**) Scheme of CLMP and CAR. Ig domains are indicated as loops. Disulfide bridges between β-sheets of Ig-domains are depicted by SS. (**B**) Crystal structure of the extracellular region of CAR at a 2.19 Å resolution (PDB: 3jz7A) [21,22] which was used to calculate the CLMP protein tertiary structure by Phyre2 [24]. β-strands in the N-terminal and C-terminal domain are labelled in uppercase letters. (**C**) Two symmetry-related mCAR monomers form a U-shaped dimer via their N-terminal domains. This dimer formation might also be predicted for CLMP.

## 4. Cellular Defects Induced by Mutations in the CLMP Gene—A Mouse Knockout Model That Phenocopies CSBS

To determine how the human *CLMP* mutations influence cell function, it is first necessary to pinpoint which intestinal cell types express CLMP during embryonic and early postnatal development. In the murine intestine, CLMP-encoding mRNA is expressed 4172-fold more strongly in the smooth muscle layer than in the villi layer. In situ hybridization and immunohistochemistry, including results from a database (www.genepaint.org, accessed on 20 December 2022), support this observation, suggesting that CLMP might exert its primary function in the smooth muscle layer of the intestine [15].

To probe the function of the cell adhesion protein CLMP, a global *Clmp* knockout (*Clmp*^−/−^) mouse model was generated [15]. This model exhibits phenotypic similarities to CSBS patients with mutations in the human *CLMP* gene, including malnutrition, lack of weight gain, intestinal malrotation, and a high mortality rate at early postnatal stages. Only about 25% of *Clmp*-deficient mice reached postnatal day 25, similar to the poor prognosis of human patients. A major discrepancy between human patients and the *Clmp*^−/−^ model is the length of the intestine in comparison to the wild-type counterpart. Whereas in human patients the small bowel fails to elongate, the intestine in the *Clmp*^−/−^ mouse is of normal length relative to body size. This phenotype has also been observed in zebrafish in which *clmp* was knocked down with morpholinos [11,15].

The *Clmp*^−/−^ mouse lacked coordinated waves of contraction in the intestine, which are essential for effective nutrient transport. This corresponded to the diminished transport of milk and meconium through the intestine. Importantly, a reduction in chyme transport was observed that corresponded to uncoordinated calcium signaling via gap junctions in the intestine. Indeed, gap junctions were severely reduced in the circular smooth muscle layer, as revealed by electron microscopy. The amounts of connexin-43 and 45—the building blocks of gap junctions in the circular smooth muscle layer—were also severely reduced at the protein level in the CLMP-deficient mouse (Figure 3). However, goblet cells were not reduced in number in the *Clmp*^−/−^ intestine, in contrast to what has been described for zebrafish *clmp* mutants [11,15]. Enteric neurons of the gastrointestinal tract are also known to modulate intestinal motility [25,26]; however, histological and physiological investigations of the *Clmp*^−/−^ mouse did not reveal any deficits in neuronal organization or communication via acetylcholine receptors in the intestine.

## 5. Mutations in the *CLMP* Gene Induce Defects in Other Tissues and Organs, including Impaired Peristalsis in the Ureters of Mice

In addition to the intestine, CLMP is also found in other tissues and cells, including the brain [27], cardiac fibroblasts of the heart [28], adipocytes [17], the ureter [15], colon and schwannoma cancer cells [29,30], the skin [31], granulosa and theca cells of the ovary [32], and endothelial and immune cells [33]. Interestingly, the expression of CLMP is reduced in Sertoli cells by the pro-inflammatory cytokine TNFα and enhanced in human pluripotent stem cells by the type I interferon signaling pathway [34,35]. As a consequence, distinct *CLMP* gene mutations could induce alterations in tissues and organs other than the intestine in CSBS patients, which should be considered when diagnosing these individuals.

### 5.1. Smooth Muscle Cells of the Urogenital System

Ureteropelvic junction obstructions were identified in two human patients with *CLMP* mutations, possibly due to impaired ureteral peristalsis [5]. This phenotype has not been observed in other patients so far. Either the penetrance of specific *CLMP* gene mutations may be variable, or ureteral deficits could develop at more mature stages in humans and therefore might have been overlooked due to the early lethality of many patients. Supportive evidence of deficits in the urogenital system comes from studies of CLMP-deficient mice. Mice that reach the post-weaning phase develop a severe bilateral hydronephrosis due to defects in the contraction of the ureter [15]. It develops perinatally and aggravates rapidly by the dilation of the renal pelvis and the loss of renal parenchyma. A functional, rather than physical, obstruction of the ureter was found in these mice. Cultures of embryonic ureter exhibited impaired peristalsis, which correlated with uncoordinated calcium waves. Therefore, hydronephrosis in CLMP-deficient mice is driven by a lack of urine transport from the kidney to the bladder. The molecular analysis of ureteral tissue revealed the reduced expression of connexin-43 in the smooth muscle layer of the ureter, similar to what has been found in the intestine [15]. Therefore, reduced gap junction-mediated coupling between smooth muscle cells of the ureter is the cause of this impairment.

Smooth muscle cells in general are located in the walls of hollow visceral organs, including arteries, veins, and lymphatic vessels [36]. It is currently not known whether CLMP is also expressed in these cells in the vascular system. A detailed analysis of the expression of *Clmp*-encoding mRNA in these cell types might uncover a function of CLMP in the smooth muscle cells of blood vessels as well. This would be of great interest, since it cannot be excluded that a putative reduced contraction of smooth muscle cells in the circulatory system could negatively affect the development and elongation of the intestine.

### 5.2. Cardiac Fibroblasts of the Heart

It was recently reported that CLMP is highly upregulated in murine cardiac fibroblasts—but less so in myogenic cells—in the injured zone of the heart after a myocardial infarction [28]. In this same study, mice expressing only 50% of wild-type CLMP levels exhibited more severe heart dysfunction than their wild-type peers after myocardial infarction, as demonstrated by echocardiography. Consequently, the mutants exhibited larger fibrotic areas and an increased inflammatory response in comparison to wild-type mice after infarction was detected. Overall, these data suggest that CLMP has a protective role after heart injury by modulating inflammatory cell death. Whether CLMP in cardiac fibroblasts is functionally linked to connexin43 or 45, which are essential for heart function by promoting cell-cell communication [37,38,39,40], must be addressed in future studies.

### 5.3. Development of Adipocytes and of Obesity

CLMP (ACAM) was also identified in a screen for genes upregulated in visceral adipose tissue in a rat model of type 2 diabetes [17], and is abundantly expressed on the cell surface of mature adipocytes. In humans, the mRNA levels of CLMP in subdermal adipose tissue are positively correlated with BMI. Surprisingly, transgenic mice in which CLMP (ACAM) was strongly overexpressed showed reduced body weight gain in comparison to their wild-type counterparts when the mice were fed a high-fat, high-sugar chow [41]. The authors of the study concluded that the overexpression of CLMP (ACAM) in mice protects from obesity and diabetes. A reduced accumulation of white and brown adipose tissues was consistently detected in CLMP (ACAM) overexpressing mice.

Interestingly, intercellular communication through gap junctions is required for the mitotic clonal expansion of adipocytes and their differentiation [42,43,44]. This suggests that CLMP could influence adipocyte function by modulating the localization and expression of connexins in this cell type, as it does in smooth muscle cells, although this must be determined experimentally.

### 5.4. Brain

CLMP has been identified in several brain areas, including the hippocampus where it is strongly expressed. It reaches peak expression in the mouse brain at early postnatal stages [15,17,27]. Importantly, one CSBS patient with a mutation in the *CLMP* gene has been described who had very mild mental retardation [5], suggesting that CLMP might be important for the development or the function of the brain. Such deficits have not been reported for other CSBS patients.

More recently, Jang et al., studied the effect of CLMP ablation on the mouse brain [27]. Unfortunately, these authors used a global knockout of *Clmp*, which is characterized by the severe phenotype described earlier, including malnutrition and hydronephrosis. While the analysis of molecular interactions between CLMP and AMPA receptor subunits or with kainate receptor subunits are not affected by these drawbacks, their electrophysiological measurements and behavioral observations should be interpreted with great caution and be considered as preliminary. These results include measurements of synaptic transmission in brain slices, which showed increased miniature excitatory post-synaptic currents in CA3 pyramidal neurons in the absence of CLMP. Furthermore, their observations that *Clmp*^−/−^ mice displayed increased susceptibility to kainate-induced seizure and enhanced object recognition memory must be reconsidered.

In conclusion, further research should be performed using mice with localized ablation of CLMP in the brain tissues to circumvent the intestinal and ureteral contributions. This would better address whether CLMP is a synaptic cell adhesion molecule involved in the negative regulation of synaptic transmission mediated by AMPA receptor subunits and kainate receptor subunits. In addition, possible links between gap junctions and chemical synapses must be taken into account, as CLMP can impact the expression and localization of connexins [45].

## 6. Functional and Structural Similarities between CLMP and Members of the CAR Family

Together with CAR, BT-IgSF, and ESAM, CLMP forms a small subfamily within the immunoglobulin superfamily called the CAR family. These proteins all contain a similar overall domain structure including a V- and C2-type Ig-like domain in the extracellular region [46]. They are also highly related at the amino acid level. A hallmark of CAR-like proteins is the presence of PDZ (PSD-95/Discs-large/ZO-1) recognition motifs at their C-terminal ends, indicating that they bind to scaffolding proteins. Such scaffolding proteins have been defined for CAR, BT-IgSF, and ESAM, and include ZO-1, PSD95, and MUPP1 [46]. A scaffolding protein has yet to be identified for CLMP, although CLMP was found to co-localize with ZO-1 in polarized epithelial or intestinal cells [11,16]. CLMP terminates in the amino acid residues’ QTV in mice and humans—a sequence that does not fully match the consensus sequence for binding to PDZ-containing proteins. Besides structural relationships, functional similarities also exist between CLMP, CAR, and BT-IgSF. Similar to CAR and BT-IgSF, CLMP can mediate the homophilic cell aggregation of transfected cell types in culture [16,17]. Of great functional importance, the ablation of any of these proteins in mice disturbs gap junction-mediated coupling. Such deficits have been found in embryonic and adult cardiomyocytes lacking CAR, and in astrocytes in the absence of BT-IgSF in the brain [46,47,48,49,50,51].

Gap junctions consist of an array of densely packed transmembrane channels built by connexins [45,52,53]. They are widely distributed and enable direct cell-to-cell communication to coordinate multiple cellular functions, including coordinating the beating of the heart, the contraction of the intestine, and the harmonizing of neuronal activity in the brain. Gap junctions are often found near tight junctions, and the scaffolding protein ZO-1 is known to bind to several connexins [54,55,56,57]. This could explain why CAR-like proteins have previously been defined as tight junction proteins in the literature. However, this likely does not reflect their true cellular function, as the knockout of individual CAR family members does not affect tight junction function and integrity [49]. However, it remains to be established by genetic experiments how CAR-like proteins exert their function on connexins.

In contrast to CLMP, the subfamily members CAR, BT-IgSF, and ESAM are not yet associated with a human mendelian disorder.

## 7. Outlook—A Putative Link between CLMP and the Actin Cytoskeleton

Several genes have been described in mouse models that interfere with the elongation of the intestine. Most of these genes are linked to the contractility of circular smooth muscle cells [58]. To better understand the role of CLMP in CSBS, it would be of interest to investigate a link between *CLMP* and some of these genes. In 2013, Van der Werf, et al. [59] presented data for three males from two families who exhibited phenotypes that strongly resembled CSBS [60,61]. Only the males were affected in these families, pointing to an X-linked gene. A two-base deletion in the second exon of the X-linked gene *FLNA* (filamin A) was detected in these three patients, suggesting that, in addition to *CLMP*, mutations in *FLNA* may cause CSBS. The two-base deletion resulted in a frameshift and produced a premature stop codon. Filamin A is an intracellular protein with an actin-bundling function and mutations in *FLNA* are associated with a wide spectrum of disorders including multiple organ anomalies [62,63]. In the intestine, filamin A is implicated in peristalsis and localized to the smooth muscle cell layer, where CLMP is also primarily expressed. Based on the phenotypic similarities, Van der Werf, et al., speculated that CLMP and filamin A might act in the same protein network. However, without further molecular evidence—for example, direct binding of the cytoplasmic segment of CLMP to filamin A or an interaction partner of filamin A—the suggestion of a link between CLMP and filamin A or between CLMP and the actin cytoskeleton remains speculative. Indeed, filamin A and CLMP could be part of two independent protein networks. Another link between CLMP and the cytoskeleton comes from the study of adipocytes. 3T3 cells can be induced to differentiate into adipocytes. In these cells, CLMP (ACAM) co-localizes with myosin IIA and γ-actin [41]. Therefore, in addition to influencing the expression of connexin-43 and 45, CLMP may modulate cytoskeletal organization and function (Figure 4), although this has yet to be tested. Future studies of CLMP-deficient cells are required to establish a direct or indirect link between CLMP and the cytoskeleton.

Several additional paths of study would help elucidate the function of CLMP and better define its role in CSBS. Mechanistic studies addressing the discrepancy in intestinal length observed between humans and mice in *CLMP* mutants could improve our understanding of intestinal elongation during development in general. It is unclear why the *Clmp*^−/−^ mouse model exhibits normal intestinal length while the intestine is severely shortened in human patients. Whether the impaired peristalsis observed upon the ablation of *Clmp* correlates with the process of intestinal elongation in humans is not yet known. It is conceivable that peristalsis is less important for intestinal elongation in mice or the process of elongation, rotation, and herniation might be interrupted or delayed in patients with CSBS [58].

In addition to an analysis of the direct or indirect interactions between CLMP and the cytoskeleton, future studies could focus on additional molecular changes that occur in the *Clmp*^−/−^ murine intestine. An analysis of the expression of gap junctions and connexin-43 and 45 at the protein level in human CSBS patients is essential. Developing mouse models of the specific human mutations that cause CSBS will be another important area of research to further our understanding of CLMP’s molecular activities. Are specific mutant forms of the protein stuck within intracellular compartments or degraded? Peristalsis of the intestine—a main defect identified in the *Clmp*^−/−^ mouse model that is essential for mixing the intraluminal material to foster the digestion and absorption of nutrients—might be carefully studied in human patients by real-time ultrasound. This work could help to improve the prognosis of CSBS patients, together with surgical interventions through intestinal lengthening or pharmaceutical therapies.

Overall, we conclude from the mouse studies that a lack of coordinated peristalsis due to the diminished expression of gap junctions in the circular smooth muscle layer is a major phenotype of CSBS. This affects cell-cell communication and coordinated calcium signaling, and consequently results in malnutrition and severe delay to thrive. Stimulation of the peristalsis of the circular smooth muscle might be a promising method for stimulating intestinal length in humans.

## Figures and Tables

**Figure 1 ijms-24-05719-f001:**
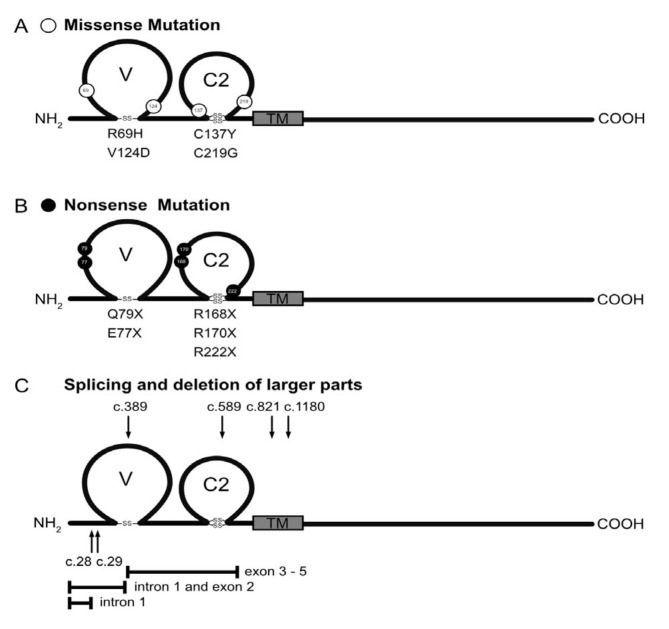
Overview of mutations in the human *CLMP* gene that coincide with CSBS. Depicted here are schematic representations of CLMP that show the localization of (**A**) missense mutations, (**B**) nonsense and frameshift mutations, and (**C**) mutations leading to splicing errors and deletions of larger parts of the gene. Numbering starts with position 1 at the amino-terminal methionine of the unprocessed polypeptide. The single letter code for amino acids is used, with “X” indicating translation stop. All frameshift mutations lead to the introduction of novel amino acid residues before the premature translational stop. Splicing mutations involve intronic sequences and therefore their positions are only roughly indicated by arrows above and below the schematic representation of CLMP. Their precise effect on the protein structure may vary and should be tested experimentally. The deletion of larger parts of the *CLMP* gene are indicated by bars below the scheme (deletion of exon 3 to 5; deletion in intron 1 together with exon 2 or deletion in intron 1). Different sizes of deletions, encompassing exons 3 to 5, of 3164 bp, 1629 bp, or 4227 bp have been described.

**Figure 3 ijms-24-05719-f003:**
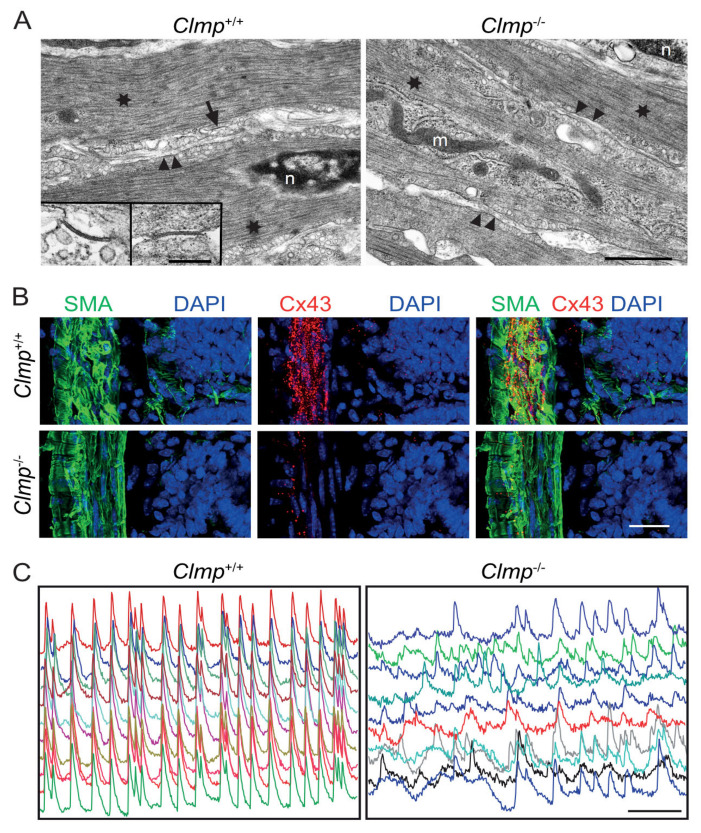
CLMP-deficient mice exhibit a reduction in gap junctions and the expression of connexin-43 in the intestinal smooth muscle layer. (**A**) Electron microscopy reveals a severe reduction in gap junctions in the *Clmp*^−/−^. The arrow denotes a gap junction in the wild-type smooth muscle layer, with enlargements shown in the insets. Scale bar, 1 µm and 200 nm (insert). Arrowheads, plasma membrane; m, mitochondria; n, nucleus; stars, actin-myosin filaments. (**B**) Immunohistochemistry for connexin43 (Cx43) shows a reduction in Cx43 clusters in the circular smooth muscle layer of the intestine in the CLMP-deficient mouse. Scale bar, 20 µm. (**C**) The absence of CLMP induces uncoordinated calcium waves in the smooth muscle layer of the intestine, which are regulated by gap junctions (modified and reproduced from [15]). Different regions of interest are indicated by different colors. Time scale bar, 100 s.

**Figure 4 ijms-24-05719-f004:**
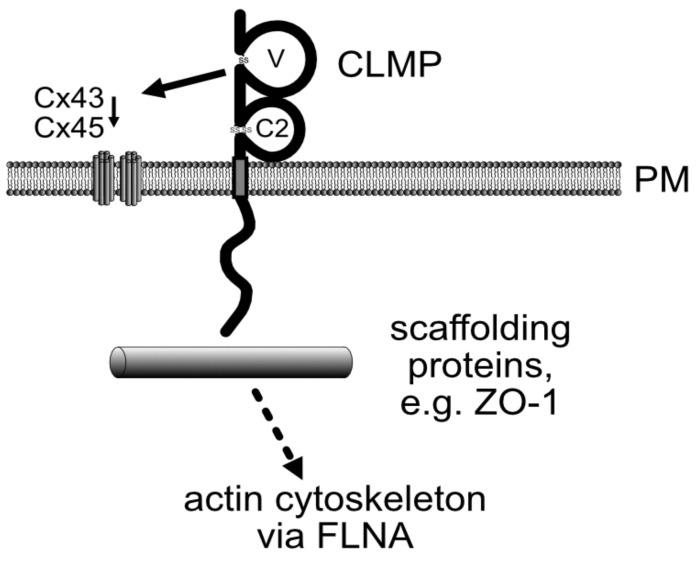
Scheme showing hypothetical molecular functions of CLMP in smooth muscle cells of the murine intestine. The absence of CLMP results in a severe reduction of connexin-43 and 45-containing gap junctions in the circular smooth muscle layer. As discussed, CLMP might also interact—directly or indirectly—with the cytoskeleton. This might also include filamin A—an actin bundling protein. The binding of the cytoplasmic segment of CLMP to a scaffolding protein such as ZO-1 or to the actin cytoskeleton must be tested experimentally.

## Data Availability

Not applicable.

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
