# Peer review of "The IgSF Cell Adhesion Protein CLMP and Congenital Short Bowel Syndrome (CSBS)"

_ijms, 2023, doi:10.3390/ijms24065719_

Round 1

Reviewer 1 Report

Very interesting and well written review paper. One thing that, according to the Reviewer, is missing is the Introduction, in which the authors would describe in a more general way the need to address such a topic, as well as the Summary of the research results and mechanisms presented in the work, and possibly their usefulness in the clinical aspect.

Minor comments:

The description of Figure 2 lacks information whether the presented results belong to the authors of the manuscript, whether they have already been published, etc.

Author Response

Reviewer 1

We thank for the overall positive comments on our manuscript. All changes in the manuscript are printed in Red.

One thing that, according to the Reviewer, is missing is the Introduction, in which the authors would describe in a more general way the need to address such a topic, as well as the Summary of the research results and mechanisms presented in the work, and possibly their usefulness in the clinical aspect.

We feel that there might be a slight misunderstanding. Our paragraph on the “Characteristics of CSBS” is thought to be the introduction. We have therefore entered the word “Introduction” to this heading. In this paragraph we also explain the need for this review. The major aim of our essay is to update our knowledge on the severe disease of congenital short bowel syndrome (CSBS). Currently, there is no cure available on CSBS. We hope very much that our review might contribute to intensify research on CSBS and to bring together pediatrics and basic researchers. This might foster the development of pharmacological treatments.

The description of Figure 2 lacks information whether the presented results belong to the authors of the manuscript, whether they have already been published, etc.

The original Figure 2 is now Figure 3. We have cited the corresponding literature in which the original data of this Figure were published.

Reviewer 2 Report

This is an excellent review covering the basic biology of the adhesion molecule CLMP and pathophysiology that results from mutations in the gene that encodes for CLMP.   The manuscript mostly focusses on congenital short bowel syndrome (CSBS) that results from mutations in CLMP, but also extends to other defects that occur with mutation of CLMP found in other tissue systems.  The review very nicely covers the history of CLMP, what is known about the molecular features of CLMP, and how disease mutations impact the gene/molecule.  Th authors also review what has been learnt from mouse knockout studies of CLMP.  In the summary the authors explore potential future avenues of research for this field, including highlighting critical questions that remain about the biology.

The text is well-written and easy to read, and the figures are well-designed.

Minor suggestion:  My one suggestion is to add an additional figure to section 3 (A survey of mutations in the human CLMP gene linked to CSBS and their putative 103 functional consequences).   The authors make the justified comparison of CLMP to the known structure of murine CAR, and begin to describe detailed structural features of CAR.  It would be helpful to include a picture of the CAR structure, to allow the reader to follow along.

Author Response

We thank for the overall positive evaluation of our manuscript by reviewer 2. Changes are indicated in Red.

Minor suggestion:  My one suggestion is to add an additional figure to section 3 (A survey of mutations in the human CLMP gene linked to CSBS and their putative 103 functional consequences).   The authors make the justified comparison of CLMP to the known structure of murine CAR, and begin to describe detailed structural features of CAR.  It would be helpful to include a picture of the CAR structure, to allow the reader to follow along.

We have added an additional Figure on the structure of CAR and CLMP in our revised version of the manuscript (Figure 2). It includes a scheme of both molecules, the crystal structure of CAR and the deduced structure of CLMP and the dimer that is formed by CAR and most likely also by CLMP.